# Native T2 Predicts Myocardial Inflammation Irrespective of a Patient’s Volume Status

**DOI:** 10.3390/diagnostics13132240

**Published:** 2023-06-30

**Authors:** Jan Sebastian Wolter, Julia M. Treiber, Selina Fischer, Ulrich Fischer-Rasokat, Steffen D. Kriechbaum, Andreas Rieth, Maren Weferling, Beatrice von Jeinsen, Andreas Hain, Christian W. Hamm, Till Keller, Andreas Rolf

**Affiliations:** 1Kerckhoff Heart and Thorax Center, Department of Cardiology, Benekestrasse 2-8, 61231 Bad Nauheim, Germany; j.wolter@kerckhoff-klinik.de (J.S.W.);; 2German Center for Cardiovascular Research (DZHK), Rhine-Main Partner Site, 60590 Frankfurt am Main, Germany; 3Justus-Liebig-Universität Gießen, Medicine, Medical Clinic 1, 35390 Giessen, Germany

**Keywords:** CMR, T2, PVS, myocardial inflammation

## Abstract

Myocardial inflammation and edema are major pathological features in myocarditis. Myocardial tissue water content and myocardial edema can be quantified via T2 mapping. Thus, cardiac magnetic resonance (CMR) is the noninvasive gold standard for diagnosing myocarditis. Several studies showed an impact of short-term volume changes on T2 relaxation time. Plasma volume status (PVS) is a good surrogate parameter to quantify a patient’s volume status, and it is simple to use. The aim of this study was to determine the effect of PVS on the diagnostic value of T2 relaxation time in myocardial inflammation. Between April 2017 and December 2022, patients who were indicated for cardiac CMR were included in our prospective clinical registry. Patients with myocardial inflammation and those with unremarkable findings were analyzed in the present study. A blood sample was drawn, and PVS was calculated. Patients were separated into PVS tertiles to explore a possible nonlinear dose–response relationship. Logistic regression analysis was used to determine whether T2 is an independent predictor of myocardial inflammation. A total of 700 patients (47.43% female) were eligible for analysis. Of these, 551 patients were healthy (78.7%), while 149 (21.3%) showed signs of myocardial inflammation. The T2 relaxation time was elevated in patients with myocardial inflammation (40 ms [IQR 37–42 ms] vs. 38.0 ms [IQR 36–39 ms], *p* < 0.001). PVS showed no difference between the groups (−12.94 [IQR −18.4–−7.28] vs.−12.19 [IQR −18.93–−5.87], *p* = 0.384). T2 showed a clear dose–response relationship with PVS, with increasing T2 values along the PVS tertiles. In spite of this, T2 was found to be an independent marker of myocardial inflammation in logistic regression (OR T2 1.3 [95% CI 1.21–1.39], *p* < 0.001), even after adjusting for PVS (OR T2 [adj. PVS] 1.31 [95% CI 1.22–1.40], *p* < 0.001). Despite a dose–response relationship between T2 and the volume status, T2 was found to be an independent indicator of myocardial inflammation.

## 1. Introduction

Myocarditis is an important differential diagnosis of chest pain, heart rhythm disorders, and, above all, heart failure [1,2]. The incidence of myocarditis in the pre-COVID era was found to be approximately 22/100.000 in 2 population studies [3,4] It is estimated that up to 20% of myocarditis patients will develop heart failure in the course of the disease [5]. Furthermore, myocarditis frequently leads to sudden cardiac death, especially in the young [6,7]. Early diagnosis is, therefore, paramount to facilitate either antiviral or immunosuppressive therapy [8].

Cardiac magnetic resonance imaging (CMR) has become the gold standard in the noninvasive diagnosis of myocarditis because of its unique characterization of tissue and its high reproducibility [2,9]. The updated Lake Louise criteria for diagnosing myocarditis are evidence of tissue damage represented by elevated T1 or late gadolinium enhancement (LGE) and evidence of edema, preferably represented by elevated T2 times of the tissue [10]. T2 times not only confirm the diagnosis but also help in monitoring the course of the disease and inflammatory activity [10,11,12,13]. This is especially true for chronic myocarditis and its involvement in heart failure, in which elevated T2 time is the foremost criterion [13,14].

On the other hand, volume fluctuations due to heart failure decompensations are hallmarks of the disease in both acute and chronic heart failure. Recent reports suggest that volume fluctuations may also influence myocardial T1 and T2 [15]. Luetkens et al. have shown that myocardial T2 changes with dehydration and rapid fluid intake [16], and the same effect was shown by Kotecha et al. and Rankin et al. in measuring T2 before and after hemodialysis [17,18].

Thus, while the value of T2 as an inflammatory marker is well known, it remains unclear if it might be hampered by volume fluctuations. The given volume state of the patient might not be clearly visible as in the more experimental settings of hydration/dehydration or after volume filtration. We therefore sought for an easy-to-use approach reflecting volume changes that are not assessable via visible estimation.

The plasma volume status (PVS) according to Hakim’s formula, which is based on the hematocrit, is a simple and noninvasive way of estimating a patient’s volume status [19]. It was also shown to be useful in determining prognosis and was validated in the large ValHeFT study [20].

Therefore, the purpose of this study was to examine in a large cohort whether PVS influences myocardial T2 and its corresponding effect on diagnostic accuracy regarding myocardial inflammation.

## 2. Methods

### 2.1. Study Population

Our study population was drawn from a prospective CMR/biobank (BioCVI) all-comers registry of patients who underwent clinically indicated CMR at a single center (Kerckhoff Heart Center, Bad Nauheim, Germany) between April 2017 and December 2022. All patients enrolled in the registry who had a hematocrit that was measured immediately before the CMR examination were included, as this was required for PVS calculation. Clinical indications for CMR included the assessment of myocardial function, ischemia testing, viability testing, myocarditis, and differentiation of cardiomyopathy.

For this analysis, we extracted patients with confirmed myocarditis and patients with normal findings as the control group. Myocarditis was defined according to the criteria proposed by Caforio et al. in the 2013 position paper of the European Society of Cardiology (ESC) Working Group on Myocardial and Pericardial Diseases [9]:

typical clinical presentation with recent onset of chest pain, symptoms of heart failure, or arrhythmia;

myocarditis-typical tissue characterization via CMR: subepicardial or intramural LGE and elevated native T1 and T2 values;

at least one of the following features:

recent-onset arrhythmias in 12-lead ECG;

elevation of troponin above the 99th percentile.

Patients with normal findings were defined as having left ventricular ejection fraction ≥50%, left ventricular end-diastolic volume index ≤105 mL, absence of LGE and perfusion defects, and normal native T1 values.

The registry includes a standardized questionnaire about symptoms, clinical and family history, medication, a single blood sample for the BioCVI biobank, a standard CMR examination with post-processing, and a clinical routine follow-up via questionnaire or telephone interview after one year.

All patients gave their written informed consent. The registry was approved by the ethics committee of the University of Giessen.

### 2.2. CMR Acquisitions

Standard CMR was performed on a 3 Tesla scanner (Skyra, Siemens Healthineers, Erlangen, Germany) in the head-first supine position with an 18-array coil in agreement with the recommendations of the Society of Cardiovascular Magnetic Resonance (SCMR) [18]. The standard protocol includes CINE imaging, tissue characterization via T1 and T2 mapping, ECV calculation, and LGE and—where appropriate—regadenosone perfusion imaging.

### 2.3. SSFP CINE Imaging

Retrograde ECG-gated standard steady-state free precession (SSFP), TE 1.38 ms, TR 3.15 ms, flip angle 50°, bandwidth 962 Hz/px, field of view (FOV) 380 mm, voxel size 1.8 × 1.8 mm, slice thickness 8 mm, and interslice gap 2 mm, with temporal resolution 30 ms. Cine sequences were generated in 11–15 short-axis views covering the whole ventricle from base to apex and in 3 long-axis (2-, 3-, and 4-chamber) views. In cases with motion artefacts due to breathing or arrhythmia, cine images were acquired by using compressed sensing.

### 2.4. Native T1 Mapping

Modified Look-Locker sequences (MOLLI 3(2)3(2)5, Goethe CVI^®^, Frankfurt, Germany; TE 1.14 ms, TR 3.1 ms, bandwidth 108 Hz/px, FOV 350 mm, voxel size 1.4 × 1.4 × 8.0 mm, slice thickness 8 mm, adiabatic inversion pulse, 11 inversion times, and ECG-gated antegrade SSFP single-shot readout with 50° flip angle) were acquired in 3 short-axis slices from base to apex following 5 into 3 planning.

### 2.5. T2 Mapping

T2 maps were generated before the injection of contrast agent by using ECG-gated antegrade T2 prep SSFP sequences during breath-hold. The typical parameters were TE 1.34 ms, TR 4.2 ms, flip angle 12°, voxel size 1.8 × 1.8 mm, slice thickness 8 mm, and T2 prep with 0, 30, and 55 ms. Three short-axis slices were acquired in the same slice position as that of the T1 maps from base to apex.

### 2.6. Late Gadolinium Enhancement

Inversion-recovery segmented gradient-echo sequences were acquired 10 to 15 min after intravenous injection of Gd-dota (Dotarem^®^, Guerbet, Villepinte, France) (0.15 mmol/kg bodyweight) in short-axis and 2-, 3-, and 4-chamber long-axis views. The delay between contrast bolus and acquisition was recorded by the technician. The typical parameters were TE 1.97 ms, TR 3.5 ms, flip angle 20°, bandwidth 289 Hz/px, FOV 370 mm, voxel size 1.3 × 1.3 × 8.0 mm, and slice thickness 8 mm.

### 2.7. Post-Processing

All analyses were performed on a commercially available workstation (CVI42, Calgary, Canada). For volumetric measurements, automatic contour detection via CVI42 was used excluding trabecularization. Contours were carefully checked by an experienced examiner (AR and JV with level 3 CMR certification of the German Society of Cardiology). End-systolic (ESVi) and end-diastolic volumes (EDVi) were indexed to body surface area. The functional parameters ejection fraction (EF) and longitudinal (GLS), circumferential (GCS), and radial strain (GRS) were calculated.

Global T2 was defined in the midventricular septum outside of LGE regions. A region of interest (ROI) at least 2 voxels wide was drawn in an automatically generated parametric T2 map (MyoMaps, Siemens Healthineers, Forchheim, Germany) by an experienced cardiologist, and mean T2-values were calculated. To avoid partial volume effects of the blood pool, the motion correction of the native magnitude images was carefully checked, and the ROI was placed in the center of the septum as recommended by the ConSept method [21].

### 2.8. Calculation of PVS

PVS was calculated according to the Hakim formula [12] as the degree of deviation of the actual plasma volume (aPV) from the ideal plasma volume (iPV). The aPV was calculated by the hematocrit and the bodyweight [19]:$$aPV=\left[\alpha +\left(\beta \times bodyweight\;\left[kg\right]\right)\right]\times \left(1-hematocrit\right)$$
(α = 1530 in males and 864 in females; β = 41 in males and 47.9 in females).

The iPV was calculated by using the bodyweight [22]:$$iPV=\gamma \times bodyweight\;\left[kg\right]$$
(*γ* = 39 in males and 40 in females).

Finally, the PVS was calculated by using iPV and aPV [19].
$$PVS=\frac{aPV-iPV}{iPV}\times 100\%$$

A value for PVS > −13% was defined as the reference value, and PVS > −4% was found to be associated with a poor prognosis in the ValHeFT cohort [12]; both were, therefore, defined as cut-off points for the present study.

### 2.9. Statistics

All metric parameters are presented as median and interquartile range [IQR]. Categorical variables are provided as number and percentage. Differences between two groups were tested via the Wilcoxon signed-rank test. Nonlinear trends along tertiles of PVS were tested by using the Jonckheere–Terpstra test. Diagnostic prediction of the presence of inflammation was tested via univariable and multivariable logistic regression analysis presenting odds ratios. A *p*-value < 0.05 was considered to be significant. All tests were computed by using Stata 17 (Stata Corp, College Station, TX, USA).

## 3. Results

### 3.1. Baseline Characteristics and CMR Findings

From April 2017 to December 2022, 3108 patients were included in our all-comers, low-risk cohort. For this analysis, only patients with inflammatory heart disease (*n* = 149) or normal findings (*n* = 551) were used. The mean age was 56.31 years (41.46–67.87 years), and 332 were female (47.43%). Heart failure, diabetes mellitus, and reduced renal function were more prevalent in the group of patients with myocardial inflammation, whereas there was no difference in age and arterial hypertension between the groups (Table 1).

T2 relaxation time was higher in patients with myocardial inflammation (40 ms (37–42 ms) vs. 38 ms (36–39 ms), *p* < 0.001) than in healthy patients. Accordingly, cardiac biomarkers hs-cTnT and NT-proBNP, as well as inflammatory marker CRP, were also elevated in patients with myocardial inflammation (hs-troponin T 6.0 (3.4–10.2) ng/L vs. 14.6 (5.2–30.1) ng/L *p* < 0.001; NT-proBNP 95.1 (40.4–233.9) pg/mL vs. 387 (90.3–1635.5) pg/mL, *p* < 0.001; CRP 0.1 (0.1–0.4) mg/dL vs. 0.2 (0.1–0.55) mg/dL, *p* < 0.05). Ejection fraction was lower in patients with inflammatory disease (61% (56–65%) vs. 53% (37–58%), *p* < 0.001). Additional CMR data are provided in Table 2.

### 3.2. PVS and Biomarkers

PVS distribution for the whole cohort showed a normal distribution (Figure 1). There were 107 patients with values above the pathological cut-off of −4% defined in the Val-HeFT cohort [20]. Patients with PVS values above −4% had higher values of NT-proBNP (106 (41–269) pg/mL vs. 214 (78–895) pg/mL, *p* < 0.001) (Table 3). PVS was slightly but not significantly reduced in patients with myocardial inflammation (−12.94 (−18.4–−7.28) vs. −12.19 (−18.93–−5.87)). PVS was associated with NT-proBNP with a beta of 0.14 (*p* < 0.01). To test for the presence of a dose–response relationship between NT-pro-BNP and PVS, PVS was divided into tertiles (Figure 2): the first tertile was from −38.45 to −16.52, the second from −16.51 to −9.38, and the third from −9.35 to 41.5. NT-proBNP values increased from the lower to the higher tertile (73 (30–240) pg/mL vs. 127 (49–287) pg/mL vs. 151 (75–387) pg/mL, Jonkheere–Terpstra test *p* < 0.001).

### 3.3. PVS and T2 Relaxation Time

PVS was associated with T2 relaxation time (Figure 3) with a beta of 0.28 (*p* < 0.01). There was a positive dose–response relationship between PVS and T2, with T2 increasing along PVS tertiles (37 (35–39) ms vs. 39 (36–40) ms vs. 39 (37–41) ms, *p* < 0.001) (Figure 4), Jonckheere–Terpstra test *p* < 0.001). This result was still significant when only patients with myocardial inflammation were considered (37 (36–39) ms vs. 40 (38–42) ms vs. 42 (40–44) ms, Jonckheere–Terpstra *p* < 0.001).

### 3.4. PVS and T2 as Diagnostic Tool in Myocardial Inflammation

Univariable logistic regression was used to test the predictive power of T2 for the presence of myocardial inflammation, which yielded an OR of 1.30 [95% CI 1.21–1.39] (*p* < 0.001). This result was independent of PVS in multivariable logistic regression analysis (OR 1.31 [95% CI 1.22–1.41]) despite detecting a significant interaction of PVS and T2, with a higher OR of T2 in the upper PVS tertiles (OR lower tertile 1.10 [95% CI 0.98–1.25] vs. OR middle tertile 1.30 [95% CI 1.14–1.48] vs. OR upper tertile 1.57 [95% CI 1.37–1.80]; *p* < 0.001) (Figure 5).

Furthermore, we found a significant nonlinear relationship of T2 and troponin T, with the highest values of troponin T in the third tertile of T2 values (18 vs. 11 vs. 33, *p* < 0.001).

## 4. Discussion

T2 relaxation time is the most important parameter in differentiating acute or active from healed myocardial inflammation [9]. However, as T2 relaxation time varies with fluctuating volume status, it is still somewhat uncertain whether the diagnostic power of T2 might be hampered by hydration status [16,23]. Therefore, our study was designed to determine whether the diagnostic value of T2 for detecting myocardial inflammation is independent of the patients’ volume status as represented by PVS. The results show that T2 is an independent predictor of myocardial inflammation in a multivariable logistic regression model despite the existence of a nonlinear dose–response relationship between PVS and T2.

T2 relaxation times are significantly prolonged in myocardial edema, and, therefore, T2 is an important marker of myocardial inflammation [24]. Recent publications have shown that T2 can be confounded by short-term volume fluctuations, for example, by excessive water intake or hemodialysis [16,25]. Because volume overload is often an important side effect of myocarditis, especially in chronic inflammation with heart failure, this is an important issue for the diagnostic reliability of T2 [26].

The most critical challenge in estimating the effect of volume status on T2 is finding a reliable surrogate marker. Two small studies investigated the influence of volume status on T2 relaxation time [16,25], caused either by rapid fluid intake or by decongestion in acute decompensated heart failure. The study by Luetkens et al. is of special importance, as it shows that congestion, a hallmark of chronic heart failure that is often accompanied by chronic myocardial inflammation, influences T2 times [16]. Luetkens et al. performed a series of 3 CMR scans in 12 healthy people. The first scan was at baseline, the second after 12 h of food and water fasting, and the third after re-hydration [16]. The strength of this study is the experimental design with clear control of the given change in volume. However, it is not representative of heart failure patients in which small volume changes can have more extensive effects. The second study by Verbrugge et al. examined 18 consecutive patients who presented with acute decompensated heart failure and received an CMR scan at baseline and after decongestion [25]. Both studies revealed a significant difference in T2 relaxation time in volume overload. However, in a large-scale registry representative of a classical tertiary care center cohort that encompasses both healthy and diseased patients, other surrogate markers are needed.

One interesting approach is to calculate the PVS via the Hakim’s formula. PVS can easily be calculated by using body weight, sex, and hematocrit. In the Val-HeFT cohort, PVS correlated with biomarkers of intravascular filling and with plasma volume measured via the gold-standard method by using ^125^iodine-labelled human serum albumin (^125^I-HSA) [20]. Moreover, PVS has a good prognostic value in patients with heart failure as well as in patients with aortic valve disease [20,27]. A cut-off calculated in the Val-HeFT cohort showed worse prognosis in patients with values above −4% [20].

In our study, PVS values showed the same Gaussian distribution as in the Val-HeFT cohort (Figure 1). Furthermore, the correlation of PVS with NT-proBNP is in good agreement with the data provided in the Val-HeFT cohort (beta coefficient 0.15, *p* < 0.01). NT-proBNP also showed elevated values in patients with PVS above −4%, and NT-proBNP increased over PVS tertiles (Figure 2). Thus, in view of the values reported in the literature, PVS in our cohort seems to be a plausible and reliable diagnostic parameter.

Similar to the findings in the other reports, T2 values showed a nonlinear dose–response relationship with increasing T2 values along the PVS tertiles (Figure 4). The mean difference in T2 times below and above the PVS cut-off of −4% (derived from the ValHeFT prognostic data) was 1.5 ms, which is in excellent agreement with the findings of Luetkens et al., who found a prolonged T2 relaxation time of 1.7 ms. The reduction of 3.2 ms in T2 relaxation time after decongestion in the analysis by Verbrugge et al. was even larger [25]. The most likely reason for this marked discrepancy might be that in the cohort of Verbrugge et al., all patients had a massive volume overload during acute heart failure. In contrast to the inclusion criteria of Verbrugge et al., Luetkens et al. included only patients with a normal volume status.

High-sensitivity-Troponin T (hs trop-T) is an important marker of myocardial injury [8]. The findings of our study are also supported by the fact that the highest hs-trop-T values were found in the upper tertial of T2 values.

### Limitations

A limitation of our study is that it is derived from a single center. However, it uses the all-comers approach in a clinical routine, which renders a generalization of our findings more plausible. Another potential limitation is that all patients who were labeled as “healthy” initially presented with cardiac symptoms, although the CMR scan revealed no pathology with respect to function, LGE, volume, and T1. It should be noted that PVS is only a surrogate parameter for volume status and not the invasive gold standard, but its easy use made it possible to apply this parameter to a very large cohort with an adequate control group.

## 5. Conclusions

Our data confirm the findings that T2 relaxation time correlates with patients’ volume status in a larger cohort. Furthermore, the diagnostic value of elevated T2 times was found to be independent of the volume status in multivariable regression analysis, which renders T2 a very robust parameter for the diagnosis of myocardial inflammation.

## Figures and Tables

**Figure 1 diagnostics-13-02240-f001:**
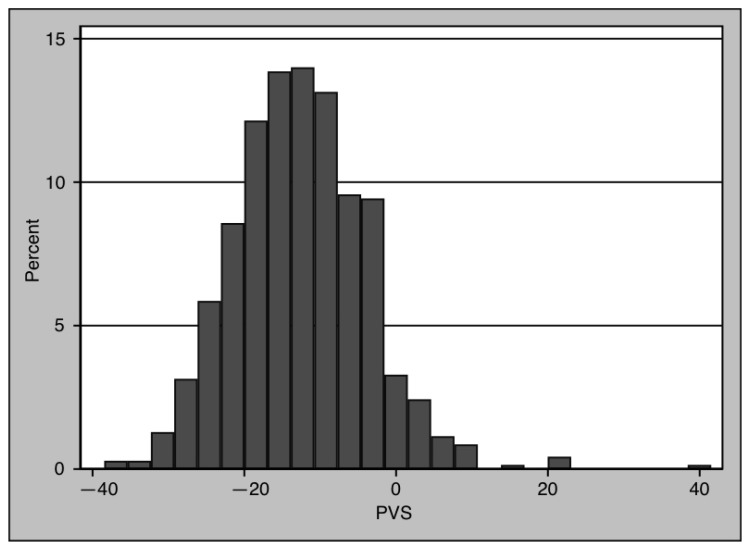
Distribution of PVS for the entire cohort. PVS showed a Gaussian distribution.

**Figure 2 diagnostics-13-02240-f002:**
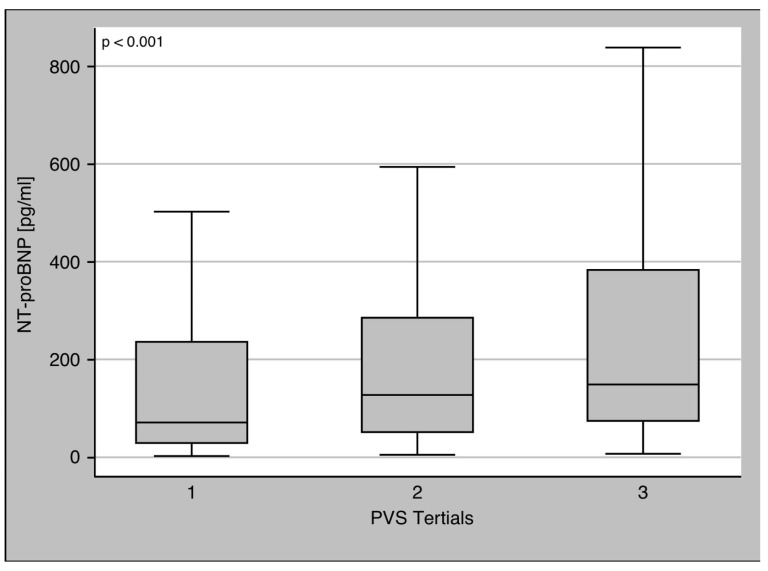
NT-proBNP in PVS tertiles. NT-proBNP showed a steady increase over all PVS tertiles (*p* < 0.001).

**Figure 3 diagnostics-13-02240-f003:**
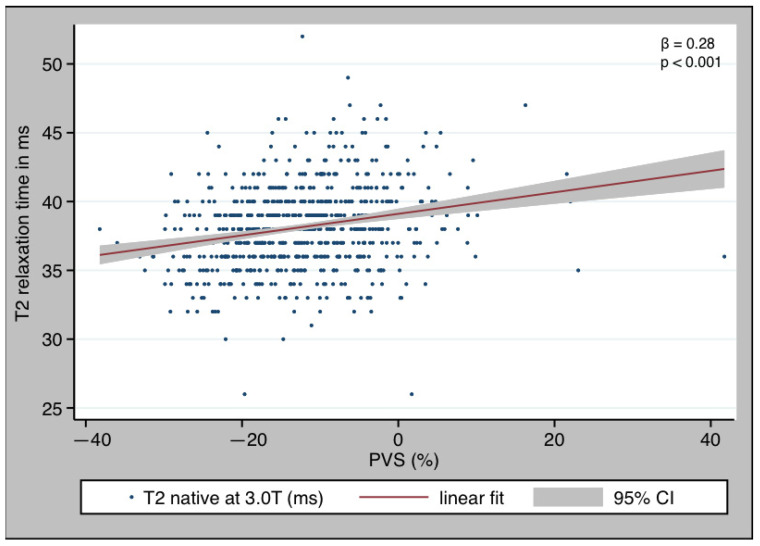
Regression analysis of PVS and T2 relaxation time. PVS and NT-proBNP showed a good correlation with a beta of 0.27 (*p* < 0.001).

**Figure 4 diagnostics-13-02240-f004:**
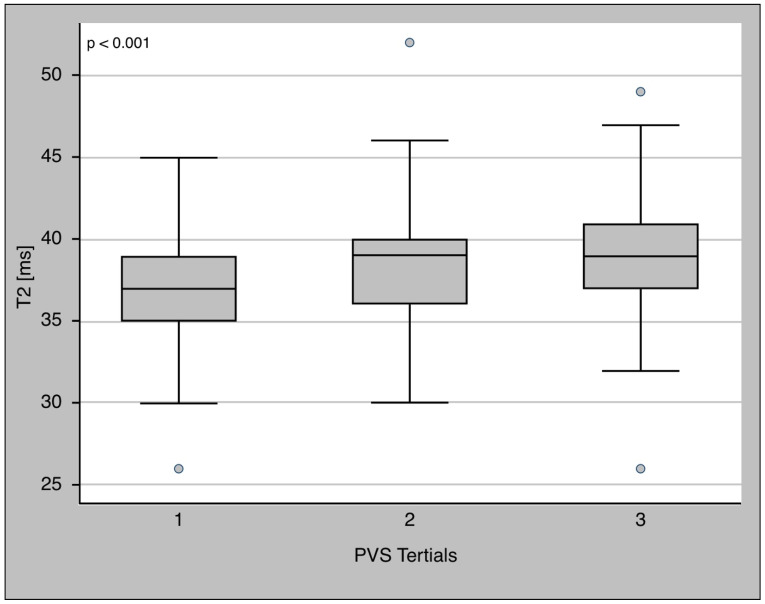
T2 relaxation time in PVS tertiles. T2 relaxation time showed a continuous increase over the four tertiles (*p* < 0.001).

**Figure 5 diagnostics-13-02240-f005:**
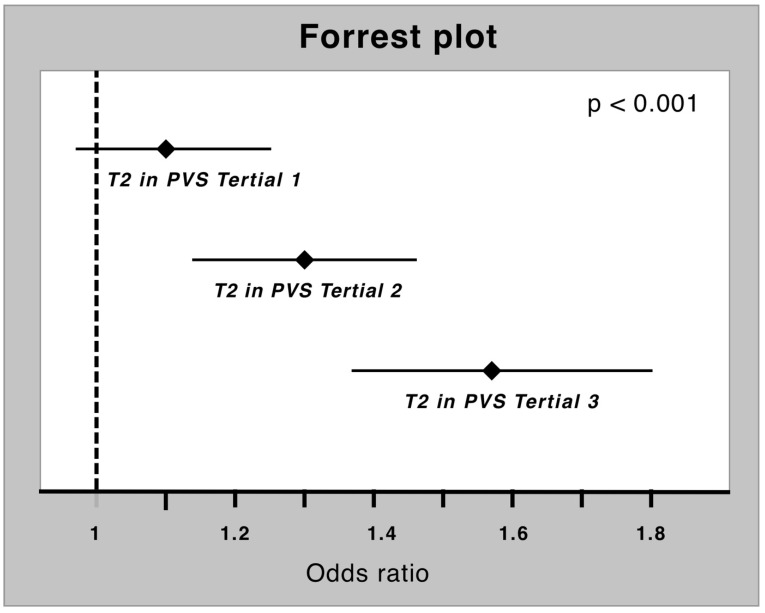
Forrest plot of T2 for prediction of myocardial inflammation in different PVS tertiles. The OR increase in every tertile (*p* for interaction <0.001).

**Table 1 diagnostics-13-02240-t001:** Baseline table for healthy individuals and patients with myocardial inflammation.

Variable	All (*n* = 700)	Healthy (*n* = 551)	Myocardial Inflammation (*n* = 149)	*p*-Value
Age [years]	56.31 (41.46–67.87)	56.69 (41.57–67.68)	55.38 (41.21–68.84)	0.782
Gender—female	332 (47.43%)	273 (49.55%)	59 (39.6%)	<0.05
CCS	71 (10.26%)	35 (6.45%)	36 (24.16%)	<0.001
Hypertonus	343 (49.21%)	272 (49.64%)	71 (47.65%)	0.668
Diabetes mellitus	78 (11.19%)	52 (9.49%)	26 (17.45%)	<0.01
Renal insufficiency	62 (8.91%)	42 (7.66%)	20 (13.51%)	<0.05
Atrial fibrillation	117 (16.79%)	87 (15.88%)	30 (20.31)	0.218
Heart failure	74 (10.69%)	18 (3.3%)	56 (38.1%)	<0.001
BMI [kg/m^2^]	25.76 (22.75–29.2)	25.71 (22.74–29.26)	25.93 (22.64–29.06)	0.786
Heart rate [bpm]	67 (60–76)	68 (60–76)	64 (56–77)	0.101
Systolic blood pressure [mmHg]	121 (115–135)	124 (117–137)	120 (110–132)	0.119
Diastolic blood pressure [mmHg]	80 (70–83)	80 (70–85)	72 (66–80)	0.003
Hematocrit [%]	42.4 (39.2–45.2)	42.4 (39.3–45)	42.1 (38.4–45.8)	0.760
Hb [mg/dl]	14.1 (13.1–15.1)	14.05 (13–15.1)	14.15 (13.35–15.33)	0.398
NT-proBNP [pg/mL]	113.5 (45.4–297.3)	95.1 (40.35–233.9)	387 (90.33–1635.5)	<0.001
hs-troponin T [ng/L]	6.85 (3.7–14)	6 (3.4–10.2)	14.6 (5.2–30.1)	<0.001
GFR [ml/min]	96.33 (80.53–116.22)	97.62 (81.93–117.08)	92.44 (73.38–109.99)	<0.01
CRP [mg/dl]	0.1 (0.1–0.4)	0.1 (0.1–0.4)	0.2 (0.1–0.55)	0.016
LV-EF [%]	59 (55–64)	61 (56–65)	53 (37–58)	<0.001
T2 relaxation time [ms]	38 (36–40)	38 (36–39)	40 (37–42)	<0.001
PVS	−12.79 (−18.53–−6.95)	−12.94 (−18.40–−7.28)	−12.19 (−18.93–−5.87)	0.384

Abbreviations: bpm—beats per minute, BMI—body mass index, CCS—chronic coronary syndrome GFR—glomerular filtration rate, LV-EF—left ventricular ejection fraction, PVS—plasma volume status.

**Table 2 diagnostics-13-02240-t002:** Comparison of CMR data.

CMR Parameter	Healthy *n* = 551 Mean (± SD)	Inflammatory Disease *n* = 149 Mean (± SD)	*p*
LV-EDVi [mL/m^2^]	75 (64–85)	93 (74–108)	<0.001
LV-ESVi [mL/m^2^]	29 (23–35)	42 (32.5–63)	<0.001
RV-EDVi [mL/m^2^]	75.5 (66–90)	85 (65.5–103)	<0.001
RV-ESVi [mL/m^2^]	36 (28–45)	44 (33–53.8)	<0.001
LV-EF [%]	61 (56–65)	53 (37–58)	<0.001
RV-EF [%]	53 (48–59)	49 (41–54)	<0.001
GLS [%]	−18.81 (−20.71–−16.78)	−16.03 (−18.72–−11.33)	<0.001
GCS [%]	−19.83 (−22.56–−17.77)	−16.23 (−18.34–−12.20)	<0.001
GRS [%]	35.71 (29.67–45.46)	26.22 (17.9–32.46)	<0.001
T1 [ms]	1105 (1075–1136)	1162 (1107–1211)	<0.001
T2 [ms]	38 (36–39)	40 (37–42)	<0.001
ECV	0.24 (0.22–0.26)	0.27 (0.23–0.31)	<0.001

**Table 3 diagnostics-13-02240-t003:** Comparison of selected parameters in patients with elevated and normal PVS values.

Parameter	PVS > −4% *n* = 107	PVS ≤ −4% *n* = 593	*p*
Inflammatory disease	31 (28.97%)	118 (28.97%)	<0.05
T2 relaxation time [ms]	39 (37–41)	38 (36–40)	<0.01
CRP [mg/dL]	0.2 (0.1–0.5)	0.1 (0.1–0.4)	0.972
NT-proBNP [pg/mL]	214 (78–895)	106 (41–269)	<0.001
hs-troponin T [ng/L]	6.4 (3.3–17.3)	6.9 (3.7–13.3)	<0.05
EF [%]	60 (54–65)	59 (55–64)	0.959

Abbreviations: LV-EF = left ventricular ejection fraction.

## Data Availability

Data available upon request at the author’s institution.

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
