# Peer review of "Native T2 Predicts Myocardial Inflammation Irrespective of a Patient’s Volume Status"

_diagnostics, 2023, doi:10.3390/diagnostics13132240_

Round 1
Reviewer 1 Report
Thank you very much for the opportunity to review this interesting work. Troponin T (TnT) is a protein forming part of the contractile apparatus of the striated muscle. The function of TnT in all types of striated muscles is the same, but cTnT is different from TnT found in skeletal muscles. Therefore, cTnT detected in plasma is a highly specific marker of myocardial damage (necrosis). High-sensitivity troponin tests, available for the past several years, detect troponin levels with a high degree of credibility (1). reference: 1. Doi: :10.20452/memory.4107 Was Troponin T assessed in the study? If so, has there been a correlation between the level of TnT and the results of the image in magnetic resonance imaging of the left ventricle?It is worth including Troponin in the discussion (1). Thank you
Author Response
(all changes made in the manuscript are highlighted in yellow)
We thank the editors and reviewers for attentively reading our manuscript. Your suggestions definitely helped to improve the quality of our manuscript. Following you find our reply to the points you raised.
Reviewer 1
Ad 1 “Was Troponin T assessed in the study? If so, has there been a correlation between the level of TnT and the results of the image in magnetic resonance imaging of the left ventricle?”
We thank the reviewer for raising this issue, as it strengthens the findings of our study. Indeed, we did measure TroponinT and found a non linear relationship with T2 and troponin. We have added these data in the results section.
Ad 2 “It is worth including Troponin in the discussion (1). Thank you”
We have added a paragraph to the discussion section and cited the corresponding reference.
We hope the changes made to the manuscript will meet the reviewers´ expectations and that the manuscript is sufficient for publication in its present form.
Reviewer 2 Report
In the manuscript, authors attempted to determine the effect of PVS on the diagnostic value of T2 relaxation time in myocardial inflammation. They collected 700 patients who got cardiac MRI. Of these, 551 patients were healthy (78.7%) while 149 (21.3%) showed signs of myocardial inflammation. after analysis, they first found that the T2 relaxation time was elevated in patients with myocardial inflammation. However, T2 relaxation time have been verified significantly associated with myocardial inflammation in many studies. Even more, as author mentioned that “The updated Lake Louise criteria for diagnosing myocarditis are evidence of tissue damage…preferably represented by elevated T2 times of the tissue”. So that the first finding was lack of innovation. Second, the result also showed that “T2 was found to be an independent marker of myocardial inflammation in logistic regression, even after adjusting for PVS.” However, myocardial T2 will changes with dehydration and rapid fluid intake, and before and after hemodialysis. Therefore, if authors wanted to prove T2 was an independent marker of myocardial inflammation, the study also need to collected patients with hemodialysis and other clinical phenotypes which will influence the T2 times. This study only compared patients with myocardial inflammation with healthy patients. This is the mainly shortcoming which authors must to be improved. Relatively small sample size was another limitation.
Author Response
(all changes made in the manuscript are highlighted in yellow)
We thank the editors and reviewers for attentively reading our manuscript. Your suggestions definitely helped to improve the quality of our manuscript. Following you find our reply to the points you raised.
Reviewer 2
Ad 1 “However, T2 relaxation time have been verified significantly associated with myocardial inflammation in many studies. […] So that the first finding was lack of innovation.”
We absolutely agree with the reviewer. It was not the purpose to repeat well known findings of T2 as marker of inflammation, it was the sole purpose to prove in a large cohort of patients, that it is not dependent on volume fluctuations. We have clarified that in the introduction.
Ad 2 “Second, the result also showed that “T2 was found to be an independent marker of myocardial inflammation in logistic regression, even after adjusting for PVS.” However, myocardial T2 will changes with dehydration and rapid fluid intake, and before and after hemodialysis. Therefore, if authors wanted to prove T2 was an independent marker of myocardial inflammation, the study also need to collected patients with hemodialysis and other clinical phenotypes which will influence the T2 times. This study only compared patients with myocardial inflammation with healthy patients. This is the mainly shortcoming which authors must to be improved.”
We thank the reviewer for raising this issue, which has to be clarified. The effect of rapid hydration and dehydration or rapid changes of volume induced by infiltration has already been demonstrated and was the motivation for our study, as we have illustrated in the introduction and discussion section. However we wanted to show that subtle changes of volume not detectable by clinical and visual assessment alone, do not influence T2 in a way, that its diagnostic value might be hampered. We have clarified this in the introduction section.
Ad 3 “Relatively small sample size was another limitation.”
Most landmark studies (for example Lurz et al. or Bohnen et al.) had less than 150 real myocarditis patients. Our cohort auf patients was derived from an all comer register covering a large spectrum of indications and diseases, we therefore truly hope our number of 149 myocarditis patients will suffice.
Round 2
Reviewer 2 Report
Authors have already revised appropriately and the manuscript is suitable for publication.